# Development and Evaluation of an Anti-Inflammatory Emulsion: Skin Penetration, Physicochemical Properties, and Fibroblast Viability Assessment

**DOI:** 10.3390/pharmaceutics17070933

**Published:** 2025-07-19

**Authors:** Jolita Stabrauskiene, Agnė Mazurkevičiūtė, Daiva Majiene, Rima Balanaskiene, Jurga Bernatoniene

**Affiliations:** 1Department of Drug Technology and Social Pharmacy, Faculty of Pharmacy, Medical Academy, Lithuanian University of Health Sciences, 44307 Kaunas, Lithuania; jolita.stabrauskiene@lsmu.lt (J.S.); daiva.majiene@lsmu.lt (D.M.); 2Institute of Pharmaceutical Technologies, Faculty of Pharmacy, Medical Academy, Lithuanian University of Health Sciences, 44307 Kaunas, Lithuania; agne.mazurkeviciute@lsmu.lt; 3Research and Development Department, Aconitum UAB, 54469 Kaunas, Lithuania; rima@aconitum.lt

**Keywords:** emulsion, menthol, capsaicin, amino acids, skin penetration, fibroblast

## Abstract

**Background/Objectives.** Chronic inflammatory skin disorders, such as atopic dermatitis and psoriasis, require safe and effective topical treatments. This study aimed to develop and evaluate a novel anti-inflammatory emulsion enriched with menthol, capsaicin, amino acids (glycine, arginine, histidine), and boswellic acid. **Methods.** Three formulations were prepared: a control (E1), a partial (E2), and a comprehensive formulation (E3). Physicochemical analyses included texture profiling, rheological behavior, pH stability, moisture content, and particle size distribution. **Results.** E3 demonstrated superior colloidal stability, optimal pH (5.75–6.25), and homogenous droplet size (<1 µm), indicating favorable dermal delivery potential. Ex vivo permeation studies revealed effective skin penetration of menthol and amino acids, with boswellic acid remaining primarily in the epidermis, suggesting localized action. Under oxidative stress conditions, E3 significantly improved fibroblast viability, indicating synergistic cytoprotective effects of combined active ingredients. While individual compounds showed limited or dose-dependent efficacy, their combination restored cell viability to near-control levels. **Conclusions.** These findings support the potential of this multi-component emulsion as a promising candidate for the topical management of inflammatory skin conditions.

## 1. Introduction

Chronic inflammatory skin diseases, such as atopic dermatitis, eczema, and psoriasis, are prevalent dermatological disorders that significantly impact patients’ quality of life and overall health [1]. These conditions are characterized by impaired skin barrier function, increased transepidermal water loss, and persistent secretion of pro-inflammatory cytokines, leading to pruritus, erythema, and alterations in epidermal structure [2]. Topical anti-inflammatory agents, including corticosteroids, calcineurin inhibitors, and nonsteroidal anti-inflammatory drugs (NSAIDs), remain the first-line treatment options. However, their prolonged use is associated with significant drawbacks, such as skin atrophy, telangiectasia, and systemic adverse effects [3]. Consequently, there is growing interest in alternative pharmaceutical formulations that optimize active compound penetration while minimizing systemic exposure [4].

Emulsions are well-recognized delivery systems in dermatological formulations due to their ability to co-deliver both lipophilic and hydrophilic agents, thereby enhancing drug stability, skin hydration, and controlled release properties [5]. In particular, the physicochemical characteristics of emulsions, such as droplet size, viscosity, pH, and stability, are key determinants of their bioavailability and clinical efficacy [6]. Emulsions with nanoscale or submicron droplets (<500 nm) have demonstrated improved skin penetration, whereas larger droplet systems may offer prolonged retention and release Menthol has been explored for its ability to provide topical analgesic effects via transient receptor potential (TRP) channel modulation, and could be considered in future combination formulations to enhance patient comfort and local anti-inflammatory effects [7]. Capsaicin, a known TRPV1 agonist, exerts anti-inflammatory and analgesic effects at low concentrations but may induce cellular stress at higher levels [8]. Amino acids such as glycine, arginine, and histidine play an essential role in maintaining the skin’s natural moisturizing factor, promoting collagen synthesis, and modulating inflammatory pathways [9]. Meanwhile, boswellic acid, an extract from Boswellia serrata, has demonstrated potent anti-inflammatory effects through inhibition of 5-lipoxygenase and NF-κB signaling; however, its highly lipophilic structure limits its diffusion through the hydrophilic layers of the skin [10].

Although these bioactives have been individually investigated, few studies have explored their combined incorporation into a single emulsion system. Most literature focuses on dual combinations or single-agent effects, which fail to capture potential synergistic interactions.

Furthermore, there is a lack of comprehensive data on the effects of emulsions on fibroblasts under oxidative stress conditions. Fibroblasts are key skin cells responsible for wound healing and collagen synthesis, yet oxidative stress can significantly impair their function [11,12].

To address these gaps, this study focuses on the development and evaluation of a multifunctional anti-inflammatory emulsion that incorporates menthol, capsaicin, a blend of amino acids (glycine, arginine, and histidine), and boswellic acid. The investigation includes physicochemical characterization, skin penetration analysis, and assessment of fibroblast viability under oxidative stress conditions. By evaluating both the individual and combined effects of these compounds, the study aims to provide insight into their potential synergistic activity and the role of emulsion properties in enhancing topical anti-inflammatory therapy.

## 2. Materials and Methods

### 2.1. Materials and Reagents

Menthol (≥99%), capsaicin (≥95%), L-glycine (≥99%), L-arginine (≥98%), L-histidine (≥98%), and boswellic acid (≥85%) were obtained from Sigma-Aldrich (St. Louis, MO, USA). Oleic acid (Ph. Eur. grade) was sourced from Honeywell Research Chemicals (Seelze, Germany).

Olive 1000 MB emulsifier (INCI: Cetearyl Olivate and Sorbitan Olivate, ECOCERT certified, origin: Italy) and the ECO preservative system (INCI: benzyl alcohol, salicylic acid, glycerin, sorbic acid) were supplied by Alexmo Cosmetics GmbH (Stuhr, Germany).

Ethanol (96%) was purchased from Vilniaus degtinė (Vilnius, Lithuania). Purified water was produced using a Millipore system (Merck, Rahway, NJ, USA).

### 2.2. Equipment and Instruments

The experimental procedures were carried out using a range of calibrated laboratory equipment to ensure the accuracy and reproducibility of the results. An analytical balance (EMB 200-3, KERN & Sohn GmbH, Balingen, Germany) was used for the precise weighing of all ingredients. A water bath (J. P. Selecta, Abrera, Spain) and a heating plate (Stuart SB160, Stone, UK) were used for controlled heating during the formulation process. Temperature measurements were monitored using a digital thermometer (LCD Digital Portable Multi-Thermometer, Changzhou, China). Mixing of formulations was performed using a laboratory mixer (Tstores, Beijing, China).

Viscosity was measured using a rotational viscometer (Fungilab ALPHA SERIES, Barcelona, Spain), while phase separation tests involved centrifugation with a high-speed centrifuge (SIGMA 3-18 KS, Osterode am Harz, Germany). The pH of the samples was determined using a digital pH meter (inoLab^®^ pH/ION 7320, WTW GmbH, Weilheim, Germany). Morphological analysis was conducted with a light microscope (Motic^®^, Motic China Group Co., Ltd., Xiamen, China). Texture properties were assessed using a texture analyzer (TA.XT. plus, Stable Micro Systems Ltd., Godalming, UK), and rheological behavior was evaluated using a rheometer (MCR102, Anton Paar GmbH, Graz, Austria).

For particle size distribution analysis, a laser diffraction particle size analyzer (Mastersizer 3000, equipped with a Hydro EV unit, Malvern Panalytical Ltd., Malvern, UK) was used. All instruments were operated following the manufacturer’s specifications and calibrated prior to use to ensure measurement reliability.

### 2.3. Emulsion Formulations

Three topical emulsion formulations were developed for this study to assess the impact of various active ingredients on skin penetration and biological activity:

E1 (Control formulation): A base emulsion composed solely of excipients, devoid of any pharmacologically active compounds.

E2 (Partial formulation): Contains two lipophilic active ingredients—menthol (2%) and capsaicin (0.075%)—incorporated into the base formulation.

E3 (Full formulation): A comprehensive emulsion that includes menthol, capsaicin, and additional bioactive compounds: glycine (2%), arginine (3%), histidine (1%), and boswellic acid (1%).

Each formulation was standardized to a final weight of 100 g and was intended for external topical application. All emulsions shared a common excipient base comprising water, oleic acid, a natural emulsifier (Olive 1000), and a preservative system consisting of an ECO-compliant preservative mixture (benzyl alcohol, salicylic acid, glycerin, and sorbic acid) (Table 1).

The concentrations of the active ingredients were selected based on previously published scientific literature and complied with the permissible limits defined by current cosmetic product regulations [2,3,4,5,6,7,8,9,10,11,12,13,14].

### 2.4. Preparation of the Emulsion Composition

The preparation of the emulsion cream involved a multi-step process involving the formulation of hydrophilic and lipophilic phases, emulsification, ultrasonic homogenization, controlled cooling, and final stabilization Figure 1.

The hydrophilic phase was prepared by dissolving the water-soluble components (L-arginine, glycine, and histidine) in distilled water under continuous stirring with a magnetic stirrer, maintaining the temperature at 30 °C. In parallel, the lipophilic phase consisted of lipophilic agents such as capsaicin and oleic acid combined with an emulsifying agent (Olive 1000). Due to its solubility characteristics, menthol and capsaicin were pre-dissolved in a minimal amount of 96% of ethanol and incorporated into the formulation after the homogenization step.

Both the aqueous and oil phases were heated separately in a water bath to 65 °C to ensure complete solubilization and reduce viscosity prior to emulsification. The lipophilic phase was then gradually added to the hydrophilic phase under continuous mechanical stirring, facilitating primary emulsification.

Subsequently, the emulsion underwent ultrasonic homogenization at 20 kHz for 5 min at room temperature (25 ± 5 °C) to reduce droplet size and enhance the system’s colloidal stability.

Following homogenization, the emulsion was cooled to room temperature (~25 °C), after which a preservative system was incorporated. A final gentle stirring ensured uniform distribution of all ingredients. The prepared samples were stored at 25 ± 5 °C under room temperature for further stability assessments [14].

### 2.5. Emulsion Stability Testing

The colloidal stability of the developed emulsion formulations was assessed using a centrifugal stress test, simulating time-dependent phase separation under accelerated conditions. The analysis was performed using a Sigma 3-18 KS laboratory centrifuge (Sigma Laborzentrifugen GmbH, Osterode am Harz, Germany). Two mL of each sample was transferred into conical tubes and centrifuged at 6000 rmp/min for 10 min at laboratory temperature (25 ± 1 °C).

This procedure detected potential phase instability, such as creaming or coalescence, that could arise during storage. The centrifugation method for accelerated stability testing was adapted from prior validated protocols to assess phase separation behavior [15]. The stability of each formulation was evaluated by calculating the Stability Index (SI), as described in Equation (1):SI (%) = (1 − V_2_ ÷ V_1_) × 100(1)
where V_2_ is the total initial volume of the emulsion sample and V_1_ is the volume of oil phase (supernatant) after the centrifugation.

### 2.6. Texture Analysis

The mechanical and textural properties of the cream formulations were assessed using a TA.XT. plus texture analyzer (Stable Micro Systems Ltd., Godalming, Surrey, UK), following a modified version of the method described by M. Yang [16]. Cream samples were transferred into standardized containers and analyzed using a cone-shaped probe that matched the internal geometry of the vessel. The probe descended vertically into the sample at a constant rate of 2.0 mm/s and retracted upon reaching the predefined penetration depth. The system recorded force–time curves in real time, enabling the evaluation of mechanical resistance and deformation behavior.

Each formulation was tested in triplicate under identical conditions, and the mean values of the measured parameters were used for statistical analysis.

Textural attributes, including firmness, stickiness, spreadability, and adhesiveness, were extracted from the force–time curves. Firmness was defined as the peak positive force (N), representing the sample’s resistance to deformation during compression. Stickiness was determined by the peak negative force (N), indicating the maximum pulling force required to separate the probe from the sample.

The spreadability of the cream was expressed as the positive area under the force–time curve (N·s), corresponding to the total energy required to deform the sample. Adhesiveness was calculated as the absolute value of the negative area under the curve (N·s), representing the attractive interactions between the sample surface and the probe.

This method provided a robust quantitative assessment of the emulsions’ mechanical behavior, offering insights into product performance characteristics such as ease of application, tactile sensation, and consumer acceptability.

### 2.7. Determination of pH Values

The pH of the emulsions was measured using a Winlab^®^ Data Line pH meter (Windaus Labortechnik, Clausthal-Zellerfeld, Germany) following procedures commonly applied for semi-solid topical formulations [17]. A 0.5 g sample was dispersed in 50 mL of distilled water and mildly heated to enhance solubilization. The pH measurement was carried out after filtration through standard paper filters and cooling to room temperature. The electrode was rinsed with ethanol and equilibrated in distilled water before each measurement. Care was taken to ensure the electrode tip was fully submerged in the test sample. Measurements were repeated three times for each sample, and the mean value was calculated. For each emulsion sample, pH was measured in triplicate, and the mean value was reported.

### 2.8. Rheological Characterization

The rheological properties of the emulsions were evaluated using an MCR102 rotational rheometer (Anton Paar GmbH, Graz, Austria) with a cone-plate measuring geometry, as described in previous studies on emulsion-based systems [18]. Temperature control was achieved using a Peltier system. The test was conducted in the linear viscoelastic range under the following conditions: angular frequency of 10 rad/s, deformation of 0.2%, and a temperature ramp from 40 °C to 0 °C at a 1 °C/min rate. The complex viscosity (η*) was used as the primary rheological parameter to characterize formulation behavior across the temperature range. Rheological measurements were performed in triplicate for each emulsion sample, and the results were reported as mean ± standard deviation.

### 2.9. Particle Size Analysis

The particle size distribution of the emulsions was determined by laser diffraction using a Mastersizer 3000 system (Malvern Panalytical Ltd., Malvern, UK) equipped with a Hydro EV unit following established guidelines for emulsion systems [19]. Distilled water was used as the dispersant. The sample volume was adjusted to achieve an obscuration range between 9% and 11%. The pump speed was maintained at 2400 rpm. Measurements were carried out five times per sample, and the average was calculated. Particle size was expressed using D10, D50, and D90 percentile values to indicate uniformity and dispersion range.

### 2.10. Determination of Moisture Content

Moisture content was assessed as a critical quality parameter influencing emulsion texture, stability, and active compound efficacy. Moisture content was measured using a Kern DBS60-3 moisture analyzer, based on standard gravimetric analysis techniques for creams and emulsions [20]. Approximately 2.0 ± 0.05 g of each sample was evenly spread on the measurement plate and analyzed in triplicate. The results were averaged to determine the mean moisture content for each formulation, providing insight into the water retention capacity and structural formulation differences.

### 2.11. Ex Vivo Skin Permeation Studies of L-Arginine, Glycine, Histidine, Capsaicin, Menthol, and Boswellic Acid

Ex vivo skin permeation experiments were conducted to evaluate the transdermal delivery efficiency of the active ingredients using abdominal skin samples obtained from Caucasian female donors. Samples were sourced from the Clinic of Plastic and Reconstructive Surgery at the Lithuanian University of Health Sciences, and ethical approval was obtained (Protocol No. BE-2-42, 2023-05-03).

Skin samples were stored at −20 °C for no longer than six months prior to use. Permeation experiments were conducted using Bronaugh-type flow-through diffusion cells with a diffusion area of 0.64 cm^2^. An infinite dose (~0.5 g) of the emulsion was applied to the epidermal side of the skin and covered with aluminum foil to prevent photodegradation. After 24 h of exposure, the residual formulation was gently removed, and the skin surface was rinsed with distilled water.

A dry heat separation technique separated the epidermal and dermal layers. The epidermal side of the skin was pressed against a heated metallic surface (~60 °C), enabling layer detachment. The separated epidermis and dermis were transferred to individual tubes, each containing 1 mL of 96% ethanol, and subjected to ultrasonic extraction for 30 min using an ultrasonic bath (DT 156, Bandelin electronic GmbH & Co. KG, Berlin, Germany) [21].

### 2.12. Quantification of Active Compound Penetration via LC-MS/MS

Quantitative analysis of the penetrated actives in skin layers was performed using a Waters Acquity H-class liquid chromatograph coupled with a Xevo TQD tandem mass spectrometer. Chromatographic separation was achieved using a Waters BEH Amide column (150 × 2.1 mm, 1.7 µm) for amino acids. Lipophilic components (menthol, capsaicin, boswellic acid) were separated using a Waters BEH C18 column (100 × 2.1 mm, 1.7 µm). No sample derivatization or additional preparation was required.

Gradient elution was applied under isothermal conditions (40 °C):

Eluent A: 10 mM ammonium formate + 0.125% formic acid (for amino acids); 0.1% formic acid in water (for lipophilic analytes)

Eluent B: Acetonitrile

Mass spectrometric detection was conducted using positive and negative electrospray ionization (ESI) modes. Operational parameters included a capillary voltage of 3500 V and a source temperature of 120 °C. Nitrogen gas was employed for nebulization at a flow rate of 650 L/h. Multiple Reaction Monitoring (MRM) mode was used for quantification, targeting analyte-specific fragmentation channels [22,23].

### 2.13. Evaluation of Cytoprotective Effects of Active Compounds on Fibroblasts Under Oxidative Stress

This study’s objective was to assess the cytoprotective effects of Boswellia extract, menthol, capsaicin, L-arginine, histidine, glycine, and their mixture on fibroblast viability under conditions simulating oxidative stress [24].

### 2.14. Methodology

#### 2.14.1. Cell Culture

Human fibroblast cells were cultured in 75 cm^2^ flasks using Dulbecco’s Modified Eagle Medium (DMEM) supplemented with 10% fetal bovine serum (FBS) and 1% penicillin-streptomycin (10,000 IU/mL–10,000 µg/mL). Cultures were maintained in a humidified incubator at 37 °C with 5% CO_2_. The medium was refreshed every 2–3 days. Cells were subcultured for experimental use upon reaching 80–90% confluency.

#### 2.14.2. Sample Preparation and Treatment

After aspirating the medium, cells were washed with PBS to remove residual nutrients. Trypsin-EDTA solution (0.5%) was added (1.0 mL), and cells were incubated for 1–2 min. Detached cells were neutralized with 4 mL of culture medium, collected, and centrifuged at 1500× *g* for 5 min. The resulting pellet was resuspended in 1 mL of medium, counted, and seeded into 96-well plates at a density of 10,000 cells per well.

After 24 h of incubation, oxidative stress was induced by adding 300 µM H_2_O_2_. Experimental wells were then treated with different concentrations of the active compounds. The incubation continued for an additional 24 h.

#### 2.14.3. Cell Viability Assay

At the end of the incubation, cells were stained with propidium iodide (5 µg/mL) and Hoechst 33342 (15 µg/mL). Propidium iodide selectively stained nuclei of damaged or necrotic cells red, while Hoechst 33342 stained viable cell nuclei blue. Cell viability and morphology were assessed using fluorescence microscopy. Five random fields per well were imaged to ensure statistical robustness. Blue fluorescence indicated viable cells, whereas red fluorescence denoted membrane-compromised, non-viable cells.

### 2.15. Statistical Analysis

All experiments were conducted in triplicate, and data are presented as mean ± standard deviation (SD). Statistical comparisons between groups were performed using one-way analysis of variance (ANOVA), followed by Tukey’s post hoc test. The value of *p* < 0.05 was considered statistically significant. All graphs and data visualizations were prepared using Microsoft Excel 365 (Microsoft Corporation, Redmond, WA, USA).

## 3. Results and Discussion

### 3.1. Colloidal Stability of Emulsion-Based Creams

Figure 2 illustrates the Stability Index (SI%) values of all emulsion samples analyzed using centrifugation. The SI values ranged from 80.0% to 95.0%, indicating varying levels of colloidal stability among the formulations depending on their composition.

E1.0, E2.0, E3.0—samples immediately after preparation; E1.1, E2.1, E3.1—samples after 1 month of storage at 22 ± 2 °C.

The lowest stability indices were observed for E1 samples (E1.0 and E1.1), scoring 80%, reflecting higher susceptibility to phase separation. These samples represent control emulsions without bioactive components and demonstrate less interfacial stabilization. Notably, no improvement was observed after one month of storage, suggesting that the formulation lacks structural reinforcers.

In contrast, E2 samples (E2.0 and E2.1), which incorporated menthol and capsaicin, achieved higher SI values of 90%, indicating improved physical stability. This suggests that the presence of lipophilic active agents contributes to the integrity of the emulsion system by potentially enhancing lipid phase compatibility or acting as co-surfactants. This finding is consistent with previous work showing that emulsions without stabilizing agents tend to phase-separate over time under mechanical stress [16].

Menthol, recognized for its surfactant-like properties, likely fortifies interfacial interactions, reducing the surface tension between oil and water phases [16]. Meanwhile, capsaicin has been documented to interact favorably with lipid phases, promoting smaller droplet sizes and thus leading to a more uniform and stable emulsion structure [17].

The E3 samples (E3.0 and E3.1), which included a combination of menthol, capsaicin, amino acids (glycine, arginine, histidine), and boswellic acid, exhibited the highest stability, reaching 95% SI. The synergistic interaction among these bioactive ingredients suggests a complex network formation that likely bolsters the emulsion’s structural integrity, thus leading to a more stable cream formulation [19].

### 3.2. Texture Profile Analysis

The mechanical texture characteristics of semi-solid formulations (E1, E2, and E3) are presented in Table 2. Parameters such as spreadability, stickiness, adhesiveness, firmness, consistency, cohesiveness, and viscosity index were quantified to assess the structural and sensorial behavior of the emulsions.

Formulation E1 exhibited the highest spreadability (42.26 ± 2.1 g·s) and moderate firmness (66.71 ± 1.41 g), with low consistency (142.66 ± 1.96 g·s) and cohesiveness (−87.36 ± 1.18 g). E2 demonstrated the highest firmness (81.31 ± 1.73 g) and adhesiveness (−17.05 ± 1.5 g·s), along with increased stickiness (−43.0 ± 2.7 g), while showing reduced cohesiveness (−79.61 ± 0.59 g). The E3 formulation yielded the greatest consistency (181.17 ± 8.32 g·s) and viscosity index (−37.54 ± 2.44 g·s), with the lowest stickiness (−23.05 ± 1.2 g) and adhesiveness (−11.08 ± 1.0 g·s).

Firmness is an essential mechanical property that shows the structural integrity and rigidity of semi-solid formulations. In our study, the control sample (E1) exhibited the lowest firmness, indicating a weaker matrix, likely due to the absence of active stabilizers. In contrast, formulation E2 exhibited significantly higher firmness, supporting previous findings that lipophilic compounds such as menthol and capsaicin enhance emulsion rigidity by increasing interfacial tension and restricting droplet mobility [24].

Formulation E3 demonstrated the highest firmness, which may result from synergistic interactions between menthol, capsaicin, amino acids, and boswellic acid. Amino acids are known to contribute to interfacial cohesion via hydrogen bonding, while boswellic acid may enhance droplet packing due to its amphiphilic nature [25,26]. Spreadability, represented by the positive area under the force–time curve (g·s), indicates the energy required to deform and spread the sample. E3 maintained good spreadability, comparable to the control E1, suggesting a well-balanced consistency. Leahu’s work also reported that emulsion gels with 2–4% seed-cake content maintain acceptable spreadability, indicating that multi-component formulations can preserve ease of application while enhancing structural properties [25].

Stickiness, the peak negative force (g) during the probe’s withdrawal, represents the maximum force required to overcome adhesive interactions with the probe surface. E2 was the stickiest formulation (−43.0 g), while E1 exhibited moderate stickiness (−36.00 g), and E3 showed significantly lower stickiness (−23.05 g), indicating a less tacky and more pleasant texture for application.

Adhesiveness, expressed as the negative area under the force–time curve (g·s), quantifies the work needed to overcome adhesive forces. E2 again had the highest adhesiveness (−17.05 g·s), followed by E1 (−15.37 g·s). E3 exhibited the lowest adhesiveness (−11.08 g·s), supporting its superior tactile performance and skin feel.

The consistency values indicated notable differences among the tested formulations. The control sample E1 demonstrated the lowest mean consistency (142.66 g·s), suggesting a less cohesive internal structure and reduced mechanical strength. In contrast, the E3 formulation exhibited the highest consistency (181.17 g·s), reflecting a structurally more uniform and mechanically robust emulsion.

Cohesiveness analysis revealed that E2 had the lowest cohesiveness value (−79.61 g), which may suggest a weaker internal bonding capacity and lower structural stability under deformation. By comparison, E1 and E3 showed similar cohesiveness values (−87.36 g and −86.15 g, respectively), indicating better mechanical integrity and resilience during texture recovery.

The viscosity index, defined by the negative area under the force–time curve, was lowest in E1 (−30.60 g·s) and highest in E3 (−37.54 g·s). This implies that the E3 formulation provided the greatest resistance to tensile stress and spreading forces, further supporting its superior physical structure and product consistency.

These findings align with previous studies that have emphasized the significance of texture parameters in assessing the performance of cosmetic emulsions. For instance, Lukic et al. 2013 emphasized that parameters such as firmness, consistency, cohesiveness, and viscosity index are crucial in evaluating the quality of barrier creams. Similarly, Tafuro et al. 2024 demonstrated that rheological and texture analyses are complementary techniques helpful in predicting the cohesiveness characteristics of cosmetics [26,27].

### 3.3. Evaluation of pH Values in Emulsion-Based Creams

The pH levels of the emulsion-based cream were evaluated at three different time points: immediately after preparation, after pH adjustment, and one week post-adjustment. The results are presented in Table 3. The initial pH values of all samples were slightly alkaline, averaging around pH ~7.50 ± 0.2, which is higher than the physiological pH of human skin. Lactic acid was used to adjust the formulations to a pH of 5.75 ± 0.2, aligning closely with the natural skin range (pH 5.5–6.0), optimal for maintaining skin health and microbiome balance. The use of lactic acid as a gentle but effective pH regulator is well known in cosmetic science, as noted by Tafuro et al., 2021 [28,29].

After one week of storage, all samples maintained near-optimal pH values (between 5.75 and 6.25), confirming the stability of the acidic environment.

### 3.4. The Rheological Behavior of Emulsion Formulations

The temperature–viscosity profile of the cream formulations shown in Figure 3. All samples exhibited shear-thinning behavior, with viscosity decreasing as the temperature increased from 0 °C to 40 °C.

At 0 °C, the highest viscosity was observed for E3 (260,000 mPa·s), followed by E1 (210,000 mPa·s) and E2 (110,000 mPa·s). This indicates that the addition of amino acids and boswellic acid in E3 significantly improves structural resistance at low temperatures, likely due to enhanced molecular interactions.

At room temperature (~25 °C), the control sample E1 retained the highest viscosity (~115,000 mPa·s), while E2 and E3 were similar (~65,000–70,000 mPa·s). This suggests that menthol and capsaicin reduce viscosity by modifying internal droplet interactions, which may improve spreadability.

By 40 °C, all formulations converged to similar viscosity levels (~25,000–30,000 mPa·s), indicating full shear-thinning behavior and reduced structural resistance at higher temperatures.

However, E2 have the most stable viscosity, with only a ~2-fold decrease over the entire temperature range, compared to a ~4-fold decrease for E3 and E1. This stability indicates better rheological control, which may be useful during storage and under different temperature conditions. These results confirm that active ingredients influence temperature sensitivity, which is relevant when formulating semi-solid preparations.

The graph illustrates the temperature-dependent viscosity behavior of E1, E2, and E3 formulations over a range from 0 °C to 40 °C.

The rheological properties, evaluated by texture profile analysis (TPA), reflect the mechanical properties of emulsions under conditions simulating in situ use. Firmness, stickiness, and cohesiveness are directly related to the internal structure and viscoelastic properties of the system. The results of our texture analysis in terms of firmness and cohesion are similar to those of other scientific studies investigating the relationship between emulsion composition and structural properties. Dubuisson et al. [27] found that the texture of emulsions is influenced by their composition, especially the ratio of water and lipid phases and stabilizers. Leahu et al. [25] reported that emulsion gels made from plant-derived components exhibit texture properties such as higher hardness and elasticity when the internal structure is optimized. In the formulations we studied, menthol and emollients not only influenced the functional properties but also affected the texture uniformity. This correlates with the results of other researchers. Han et al. [17] and Mondal et al. [30] show that biologically active lipophilic compounds such as capsaicin or menthol can affect the internal matrix by changing the lipid distribution and interphase tension, which influences texture changes. The rheological profile not only shows the stability of the formulation over time but also confirms the conclusions regarding the spreading properties and consumer acceptance.

### 3.5. Particle Size Distribution

The particle size distribution of the emulsion-based creams was assessed using laser diffraction analysis, a highly sensitive and reproducible method for characterizing colloidal systems. The results revealed significant differences in droplet size among the formulations, highlighting the impact of active ingredient incorporation on emulsion structure.

As shown in Figure 4, the control formulation E1 exhibited markedly larger particle sizes compared to E2 and E3. The particle distribution curve for E1 was broader and shifted toward higher micrometer values, suggesting less efficient droplet dispersion and reduced colloidal uniformity. In contrast, both E2 and E3 demonstrated narrower, more symmetric distributions, indicating improved emulsification and structural stability, likely attributable to menthol and capsaicin, which may act as secondary emulsifiers or influence interfacial dynamics.

To further evaluate droplet size uniformity, percentile values D10, D50, and D90 were analyzed (Table 4). These parameters represent the particle diameter below which 10%, 50%, and 90% of the distribution lies.

The median particle size (D50) of E1 (7.80 μm) was 10 times larger than that of E2 (0.78 μm) and 10.5 times larger than E3 (0.74 μm).

E1 also had a wider distribution span, as shown by the immense interval between D10 and D90 values, suggesting higher polydispersity and lower emulsification efficiency.

E3 demonstrated the most uniform droplet size, with minimal standard deviation and tight clustering around the mean, indicating the high physical stability of the emulsion system.

Laser diffraction results showing differences in particle size and distribution curve sharpness among formulations.

These findings confirmed that adding lipophilic active ingredients (menthol and capsaicin) significantly improves emulsion microstructure by reducing droplet size and narrowing the size distribution range [31,32].

### 3.6. Determination of Moisture Content in Emulsions

The moisture content of the emulsion-based cream formulations was measured to assess the influence of composition on water retention and structural stability. The results are summarized in Table 5.

The results demonstrate a decreasing trend in moisture content across the samples, following the order E1 > E2 > E3. This pattern suggests that the formulation composition is critical in water retention capacity within the emulsion matrix.

The highest moisture content was observed in the E1 formulation (90.35 ± 0.16%), which is consistent with its higher proportion of water and the emulsifier’s hydrophilic characteristics (Olive 1000). The emulsion’s structure, empty of active agents, likely allowed for maximal water loading and retention.

These components may act as structural stabilizers, reducing the amount of free water and promoting stronger molecular tightness or stronger organization of emulsion interfaces. E2 showed an intermediate moisture level, suggesting that these lipophilic compounds may disrupt the emulsion matrix, reducing water retention. E3, being the most complex formulation, exhibited the lowest moisture content. The multi-component system likely altered the hydrophilic–lipophilic balance (HLB) and induced structural tightening of the emulsion interface, effectively reducing the amount of free or loosely bound water.

A recent study by Meneguello et al., in 2025 [33], evaluated the physicochemical and sensory properties of sustainable plant-based homopolymers in topical emulsions. The research found that emulsions formulated with these homopolymers maintained viscoelastic stability and offered improved skin hydration and moisture retention, highlighting the influence of formulation components on water retention and structural stability.

### 3.7. Ex Vivo Skin Permeation Study

#### 3.7.1. Control Formulation (E1)

The control emulsion E1, which contained no active compounds, exhibited negligible permeation into the epidermis and dermis. Detected trace amounts of some analytes, such as glycine (13.65 µg/mL in the dermis; 7.76 µg/mL in the epidermis), were likely due to endogenous presence in the skin or biological background interference. This confirms that the base formulation alone does not promote transdermal penetration.

#### 3.7.2. Effect of Menthol and Capsaicin in E2 Formulation

Including menthol and capsaicin in the E2 formulation significantly enhanced their permeation across the skin layers.

As shown in Table 6, menthol showed high permeation flux, confirming its deep skin penetration capabilities. Menthol’s superior flux was likely due to its lipophilic nature, low molecular weight, and ability to act as a skin penetration enhancer, facilitating its own and other molecules’ diffusion across the stratum corneum. A study by Chen L. et al. found that 1% menthol concentration did not change the structure of the stratum corneum but increased the interlayer spacing and increased fluidity [34]. Menthol also increased the spacing between lipid headgroups and reduced the number of lipid–lipid and lipid–water hydrogen bonds.

Capsaicin accumulated more in the dermis than in the epidermis, suggesting preferential dermal targeting, potentially interacting with nociceptors and pain receptors. Capsaicin exhibits good skin penetration [31], and the low flux may be due to the relatively low concentration of capsaicin in the formulations studied.

#### 3.7.3. Skin Permeation of Complete Formulation (E3)

The E3 formulation, enriched with menthol, capsaicin, glycine, arginine, histidine, and boswellic acid, was analyzed to assess synergistic effects on permeation, shown in Table 7.

Compared to E2, E3 resulted in a higher menthol flux in the epidermis; however, the difference found was not statistically significant (*p* > 0.05). This indicates that amino acids and boswellic acid may enhance epidermal diffusion, likely by modifying skin barrier dynamics. However, menthol’s fluxes are similar between E2 and E3, suggesting saturation at deeper layers.

For capsaicin, epidermal accumulation slightly increased in E3, but flux through the dermis declined (from 3.40 to 2.79 µg/cm^2^), indicating that capsaicin may stay in superficial layers when combined with hydrophilic actives.

Glycine exhibited the greatest dermal flux, indicating that this amino acid effectively penetrates the epidermis and accumulates in deeper skin layers. Arginine demonstrated the highest epidermal flux, suggesting that it tends to accumulate in the superficial layers and may function as a penetration-enhancing agent. Histidine showed intermediate permeation values compared to glycine and arginine, with substantial dermal delivery and notable epidermal flux, reflecting balanced diffusion across the skin.

In contrast, boswellic acid exhibited significantly lower permeation than the amino acids. Its concentration in the dermis was very low, and both dermal and epidermal flux values were minimal (0.78 ± 0.82 µg/cm^2^ and 1.13 ± 0.95 µg/cm^2^, respectively). Studies by other researchers show that menthol increases the penetration of boswellic acid into the skin [10,35]. However, these findings indicate that boswellic acid does not penetrate deeply into the skin but likely remains within the superficial layers, suggesting potential for localized topical activity. The poor penetration of boswellic acid into deeper skin layers can be explained by its high lipophilicity [10].

### 3.8. Effects of Boswellic Extract, Menthol, Capsaicin, L-Arginine, Histidine, Glycine, and Their Combination on Fibroblast Viability Under Pathological Conditions

#### 3.8.1. Effect of Boswellic Extract on Fibroblast Viability Under Pathological Conditions

In control wells under physiological conditions, microscopy revealed a high percentage of viable fibroblast cells—up to 98%, with less than 1% of cells undergoing necrosis and a similarly low percentage undergoing apoptosis (Figure 5A). This reflects typical morphology and viability in untreated cell cultures.

A significant decrease in viable cell count was observed upon induction of oxidative stress using 300 µM hydrogen peroxide (H_2_O_2_). Cell death predominantly occurred through apoptotic pathways, as indicated by nuclear condensation and red fluorescence signaling compromised membrane integrity (Figure 5B).

Interestingly, in the experimental group where 20 µg/mL of boswellic extract was added under oxidative stress conditions, the fibroblast morphology and cell viability appeared comparable to control samples. Microscopic images (Figure 5C) showed many viable fibroblasts, suggesting a cytoprotective effect of boswellic extract. This indicates that the extract may exert protective or restorative properties under oxidative damage by mitigating apoptosis and preserving fibroblast viability.

The results of this study demonstrated that the addition of 20 µg/mL boswellic extract under oxidative stress conditions significantly preserved fibroblast viability, showing a comparable cellular morphology and survival rate to the untreated control group. These findings suggest that boswellic extract exhibits cytoprotective activity, likely by mitigating oxidative stress-induced apoptosis and preserving membrane integrity. This observation aligns with previous studies reporting that boswellic acids—particularly 3-O-acetyl-11-keto-β-boswellic acid (AKBA)—exert potent anti-inflammatory and anti-apoptotic effects through the inhibition of nuclear factor-kappa B (NF-κB) and activation of glucocorticoid receptors [36]. *In vitro* studies have further demonstrated that boswellic acids can reduce intracellular reactive oxygen species (ROS), stabilize mitochondrial membranes, and upregulate antioxidant enzymes, including superoxide dismutase (SOD) and catalase, thereby enhancing cellular resistance to oxidative insult [3,8]. Ezra et al. reported that antioxidant-rich emulsions containing boswellic derivatives effectively protected fibroblast cultures from oxidative damage, improving cell survival under oxidative load [37]. Similar cytoprotective outcomes have been observed in studies examining Boswellia extracts applied to damaged skin models, where anti-apoptotic signaling pathways were activated and collagen synthesis preserved [6].

Collectively, these findings support the therapeutic relevance of incorporating boswellic extract into topical formulations, particularly for skin conditions characterized by chronic inflammation and oxidative damage. Given its low permeability into deeper dermal layers, as shown in our permeation study, the compound may act locally at the epidermal level—where many inflammatory skin processes initiate—offering targeted protection without significant systemic absorption. Further mechanistic investigations and in vivo studies are warranted to explore the long-term protective effects of boswellic acids, their optimal concentration ranges, and potential interactions with other active ingredients in complex formulations.

#### 3.8.2. Effect of Capsaicin on Fibroblast Viability

Fluorescence microscopy analysis revealed the dose-dependent effects of capsaicin on fibroblast viability (Figure 6). In wells treated with low concentrations of capsaicin (1–10 µg/mL) under oxidative stress conditions, a higher number of viable cells was observed than the H_2_O_2_-only stressed group (Figure 6C). However, viability remained lower than in the untreated control group, suggesting a partial protective effect of capsaicin at these concentrations.

In contrast, higher concentrations of capsaicin resulted in increased cell death, with characteristic morphological signs of necrosis observed in fibroblast nuclei (Figure 6D). This indicates that while low doses may exert cytoprotective properties, excessive capsaicin levels may induce cytotoxicity, particularly through necrotic mechanisms.

The biphasic effect of capsaicin on fibroblast viability observed in this study underscores its dual pharmacological nature. At low concentrations (1 µg/mL), capsaicin exhibited partial cytoprotective activity against oxidative stress, improving cell survival compared to the H_2_O_2_-only group. However, it did not restore viability to baseline levels, as seen with boswellic extract, suggesting limited antioxidant potential or involvement of alternative protective mechanisms. This mild benefit may be attributed to transient activation of TRPV1 receptors, which in controlled conditions has been shown to trigger pro-survival pathways such as PI3K/Akt and promote the release of growth factors [8,38]. Furthermore, low-dose capsaicin may modulate intracellular redox signaling, reducing ROS accumulation without causing cellular stress [39]. In contrast, a 20 µg/mL dose resulted in pronounced fibroblast death, predominantly via necrosis, as evidenced by nuclear disintegration and widespread red fluorescence. High-concentration capsaicin is known to cause mitochondrial dysfunction, sustained calcium influx, and oxidative imbalance, leading to irreversible membrane damage and inflammatory cell death [40,41]. This cytotoxic effect is consistent with earlier reports that capsaicin exhibits a narrow therapeutic window in cutaneous cell models [42]. Compared to boswellic extract, which uniformly preserved fibroblast viability under oxidative conditions, capsaicin’s action was concentration-dependent and less predictable. These findings emphasize the need for careful titration when including capsaicin in topical formulations. Its potential benefits—such as vasodilation, mild inflammation modulation, and local analgesia—must be balanced against its cytotoxic risk, especially in compromised or inflamed skin.

#### 3.8.3. Effect of Menthol on Fibroblast Viability

In experimental wells supplemented with 20 µg/mL of menthol, an increase in fibroblast viability was observed compared to the oxidative stress condition alone (Figure 7C). However, the number of viable cells remained lower than in the untreated control group (Figure 7A), indicating that menthol may exert a partial cytoprotective effect under oxidative stress, though not fully restoring cell viability to baseline levels.

The results of this study suggest that menthol, at a concentration of 20 µg/mL, provides limited cytoprotective effects under oxidative stress conditions. While a modest improvement in fibroblast viability was observed compared to the hydrogen peroxide group, the effect was clearly inferior to that of boswellic extract and less consistent than low-dose capsaicin. Menthol’s mechanism of action is believed to involve mild membrane-stabilizing properties and the modulation of cold-sensitive TRPM8 channels, which may influence calcium influx and cell signaling [43]. However, its impact on mitochondrial function or antioxidant enzyme activity appears minimal, which could explain its limited efficacy in restoring cellular homeostasis under strong oxidative conditions. Unlike boswellic acids, which exhibit direct anti-inflammatory and anti-apoptotic activity, or capsaicin, which activates robust intracellular pathways, menthol’s contribution may be primarily physico-chemical—altering membrane fluidity and improving topical absorption rather than offering intrinsic cellular protection [6,7]. These findings highlight menthol’s role as a supportive excipient rather than a primary cytoprotective agent. Its inclusion in dermatological formulations should be aimed at enhancing skin permeability and user sensory experience rather than serving as a standalone therapeutic compound under oxidative stress.

#### 3.8.4. Effects of L-Arginine, L-Histidine, and Glycine on Fibroblast Viability

Experimental wells treated with the amino acids L-arginine, L-histidine, and glycine showed no significant improvement in fibroblast viability under oxidative stress conditions. As seen in the representative images (Figure 8), the number and morphology of viable cells in amino acid-treated wells were comparable to those observed in wells exposed to 300 µM H_2_O_2_ alone (Figure 8B, C). These findings suggest that these amino acids did not exert notable cytoprotective effects under the applied oxidative stress conditions.

The absence of cytoprotective effects observed with L-arginine, L-histidine, and glycine suggests that, under acute oxidative stress, these amino acids do not directly enhance fibroblast viability. Although structurally involved in maintaining skin barrier integrity and collagen synthesis, their protective capacity at the cellular level may depend on concentration, formulation context, or the presence of co-factors such as antioxidants. One possible explanation is that these amino acids serve primarily as substrates or metabolic intermediates in antioxidant defense systems, rather than as immediate ROS scavengers. For instance, L-arginine is a precursor for nitric oxide (NO), which can exhibit both protective and cytotoxic roles depending on its concentration and redox environment [30]. Similarly, glycine and histidine contribute to glutathione and carnosine synthesis, respectively, but may require additional regulatory elements to activate these pathways effectively [44]. It is also plausible that amino acids exhibit a delayed or indirect effect on cell recovery by supporting protein repair or modulating post-stress regeneration, mechanisms that may not be captured within the 24 h oxidative stress window used in this study. Moreover, their known role in enhancing skin penetration of co-administered actives may explain their efficacy in the combined formulation (E3), even if they lack standalone cytoprotective properties [9].

These findings support the notion that amino acids in topical formulations should be viewed more as penetration enhancers or structural skin components, rather than direct agents for oxidative stress mitigation in dermal cells.

#### 3.8.5. Effect of the Active Ingredient Mixture on Fibroblast Viability

In the wells exposed to oxidative stress (300 µM H_2_O_2_) and simultaneously treated with the mixture of active ingredients (including boswellic extract, menthol, capsaicin, glycine, L-arginine, and histidine), fibroblast cell viability was significantly restored. As shown in the representative fluorescence microscopy image (Figure 9C), the number of viable cells was comparable to the untreated control group (Figure 9A), indicating that the combined formulation had a pronounced cytoprotective effect under oxidative conditions.

These findings suggest that while individual actives may have limited protective capabilities, their synergistic combination can effectively mitigate oxidative damage, maintaining cellular integrity and survival.

The mixture of tested active compounds and boswellic extract, at the evaluated concentrations, demonstrated effective cytoprotective properties, restoring fibroblast viability under oxidative stress conditions to levels comparable with untreated controls. Low concentrations of capsaicin showed a protective effect, whereas higher doses induced cell death via necrosis, indicating a narrow therapeutic window. Menthol significantly improved fibroblast viability under oxidative stress, although it did not fully restore viability to baseline levels. In contrast, the amino acids L-arginine, L-histidine, and glycine showed no measurable protective effects on fibroblast survival under the tested stress conditions.

The significant restoration of fibroblast viability following treatment with the active ingredient mixture highlights a synergistic interaction among the included compounds. Unlike the limited or partial effects observed when these agents were applied individually, their combination produced a cumulative cytoprotective outcome, with cell viability returning to near-control levels despite ongoing oxidative stress. Such synergy may arise from the complementary mechanisms of the actives: boswellic acids exert anti-inflammatory and anti-apoptotic effects via NF-κB inhibition; menthol modulates membrane fluidity and calcium influx; low-dose capsaicin activates pro-survival pathways through TRPV1 receptors; and amino acids, although ineffective alone, likely contribute to redox homeostasis, osmoregulation, or membrane repair in the presence of other stress-modulating agents [37,45]. This multifactorial protection suggests that oxidative damage in fibroblasts cannot be effectively mitigated by a single mechanistic route. Instead, targeting multiple cellular compartments—mitochondria, ion channels, antioxidant systems, and membrane integrity—simultaneously is required to preserve cell viability under stress. The concept of functional complementarity within the mixture is further supported by earlier findings showing that compound combinations often outperform individual agents in skin regeneration and wound healing models [5]. These results reinforce the rationale for designing multi-component topical formulations, especially when addressing complex pathophysiological conditions such as chronic inflammation and oxidative stress in the skin. However, careful balancing of concentrations remains essential to avoid potential cytotoxicity, particularly in the case of capsaicin.

## 4. Conclusions

This study demonstrated that a topically applied emulsion enriched with menthol, capsaicin, amino acids (L-arginine, glycine, histidine), and boswellic acid exhibits favorable physicochemical characteristics, enhanced skin permeation properties, and significant cytoprotective activity under oxidative stress conditions. Among the tested formulations, the comprehensive emulsion (E3) showed optimal stability, pH compatibility, and textural performance, supporting its potential suitability for dermatological application.

Skin permeation analysis revealed that amino acids and menthol contributed to improved epidermal and dermal delivery of active compounds, while boswellic acid remained largely localized in the upper skin layers, indicating targeted anti-inflammatory potential. Although individual actives had limited protective effects on fibroblast viability, their combination markedly restored cell survival under oxidative conditions, suggesting a synergistic mechanism of action.

These findings highlight the importance of rational multi-component formulation design in achieving both efficient dermal delivery and cellular protection. The results support further preclinical development of this emulsion system as a promising platform for the topical treatment of inflammation-associated skin disorders.

## Figures and Tables

**Figure 1 pharmaceutics-17-00933-f001:**
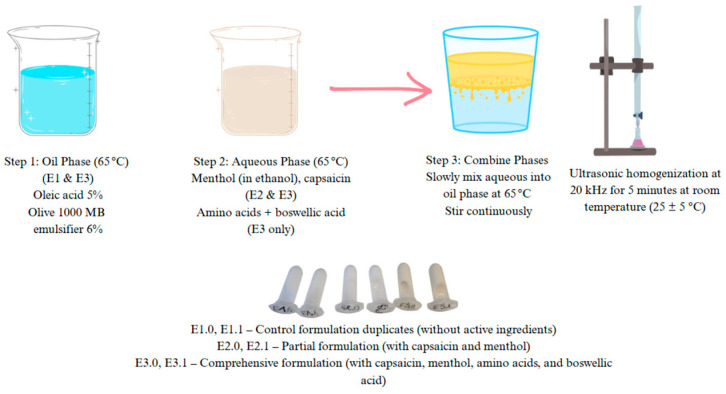
Schematic representation of the emulsion preparation process.

**Figure 2 pharmaceutics-17-00933-f002:**
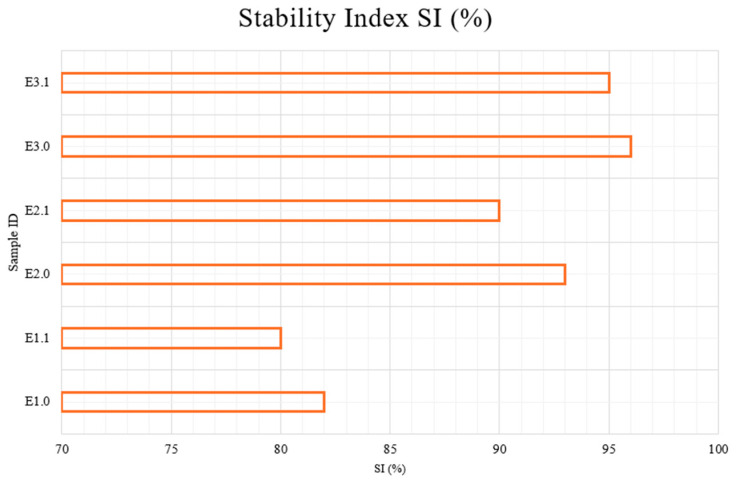
Stability Index (SI%) of cream formulations after centrifugation test.

**Figure 3 pharmaceutics-17-00933-f003:**
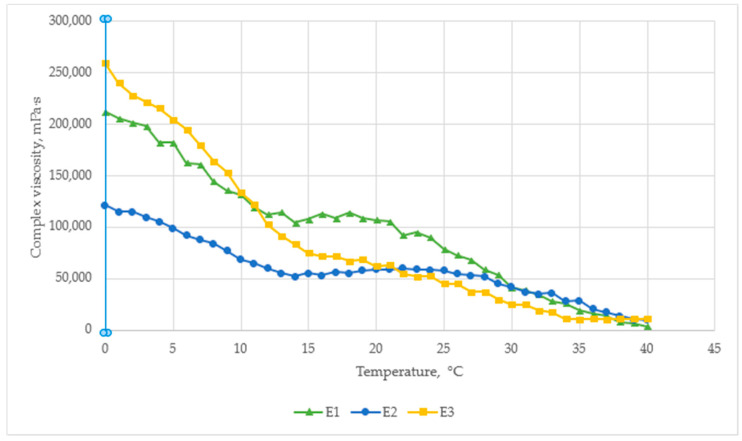
Rheogram of the tested emulsions.

**Figure 4 pharmaceutics-17-00933-f004:**
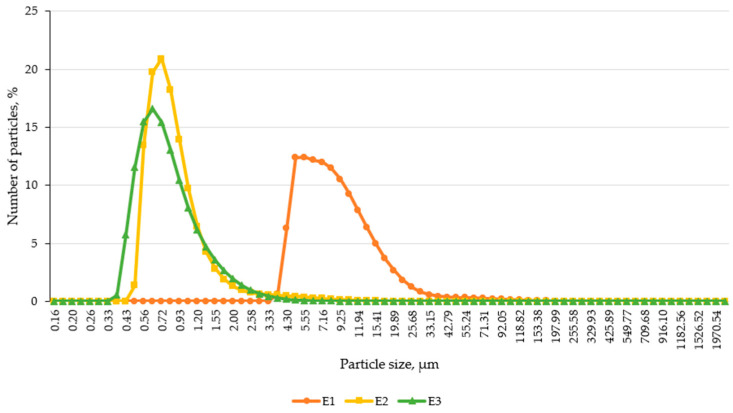
Particle size distribution profiles of emulsion samples E1, E2, and E3.

**Figure 5 pharmaceutics-17-00933-f005:**
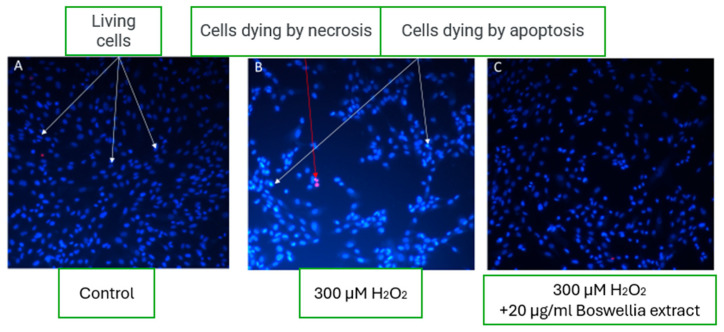
Effect of boswellic extract on fibroblast cells under oxidative stress conditions. (**A**) Control group with high viability (~98%), (**B**) oxidative stress group treated with 300 µM H_2_O_2_ showing increased apoptosis, and (**C**) oxidative stress group treated with 20 µg/mL boswellic extract demonstrating preserved cell viability comparable to control.

**Figure 6 pharmaceutics-17-00933-f006:**
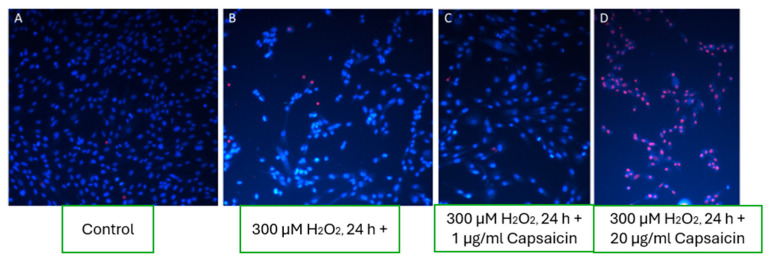
Effect of varying capsaicin concentrations on fibroblast cells under oxidative stress conditions. (**A**) Control group; (**B**) oxidative stress induced by 300 µM H_2_O_2_; (**C**) treatment with low concentrations of capsaicin (1–10 µg/mL) demonstrating partial protection; (**D**) higher concentrations of capsaicin leading to necrotic cell death.

**Figure 7 pharmaceutics-17-00933-f007:**
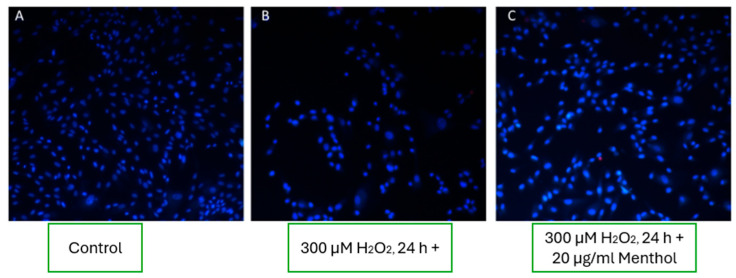
Effect of menthol on fibroblast cells under oxidative stress conditions. (**A**) Control group; (**B**) oxidative stress induced by 300 µM H_2_O_2_; (**C**) treatment with 20 µg/mL menthol under oxidative stress conditions, showing partial restoration of cell viability.

**Figure 8 pharmaceutics-17-00933-f008:**
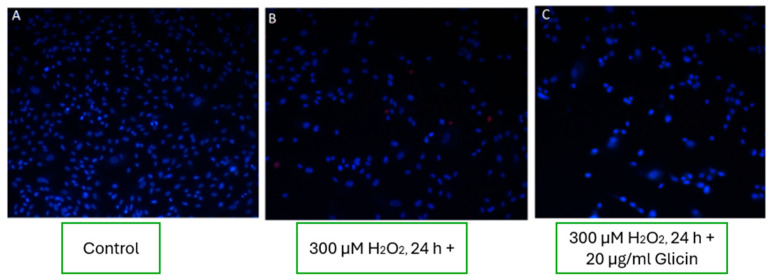
Effects of L-arginine, L-histidine, and glycine on fibroblast cells under oxidative stress conditions. (**A**) Control conditions; (**B**) oxidative stress induced by H_2_O_2_; (**C**) treatment with individual amino acids, showing no significant improvement in cell survival.

**Figure 9 pharmaceutics-17-00933-f009:**
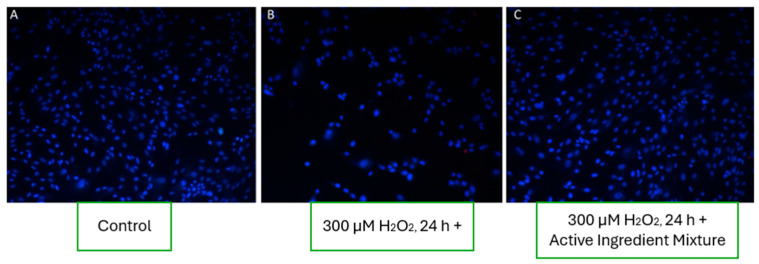
Effect of active ingredient mixture on fibroblast cells under oxidative stress conditions. (**A**) Control group; (**B**) oxidative stress group treated with 300 µM H_2_O_2_ for 24 h; (**C**) oxidative stress + active ingredient mixture, showing restored cell viability.

**Table 1 pharmaceutics-17-00933-t001:** Composition of topical emulsion formulations (per 100 g).

Component	E1 (Control)	E2	E3
Water	88.05%	86.42%	79.42%
Oleic acid	5.00%	5.00%	5.00%
Emulsifier (Olive 1000)	6.00%	6.00%	6.00%
Preservatives (ECO mix)	0.50%	0.50%	0.50%
Menthol	—	2.00%	2.00%
Capsaicin	—	0.075%	0.075%
Glycine	—	—	2.00%
Arginine	—	—	3.00%
Histidine	—	—	1.00%
Boswellic acid	—	—	1.00%

Note: Oleic acid was employed not only as an emollient but also as a skin penetration enhancer to improve dermal delivery of active compounds. Olive 1000 functioned as a natural emulsifier to stabilize the oil-in-water emulsion system [13,14].

**Table 2 pharmaceutics-17-00933-t002:** Mechanical and textural properties of emulsion-based cream formulations.

Sample ID	Spreadability (g·s)	Stickiness (g)	Adhesiveness (g·s)	Firmness (g)	Consistency (g·s)	Cohesiveness (g)	Viscosity Index (g·s)
**E1**	42.26 ± 2.1	−36.0 ± 2.5	−15.37 ± 1.4	66.71 ± 1.41	142.66 ± 1.96	−87.36 ± 1.18	−30.60 ± 1.36
**E2**	41.15 ± 1.7 *	−43.0 ± 2.7 *	−17.05 ± 1.5 *	81.31 ± 1.73 *	156.15 ± 11.64 *	−79.61 ± 0.59 *	−33.22 ± 2.98 *
**E3**	42.15 ± 1.9	−23.05 ± 1.2 *	−11.08 ± 1.0 *	88.84 ± 1.77 *	181.17 ± 8.32 *	−86.15 ± 1.42	−37.54 ± 2.44 *

Mechanical and texture properties of emulsion-based cream formulations. Values are presented as mean ± standard deviation (*n* = 3). Different superscript letters in the same column indicate significant differences * (*p* < 0.05).

**Table 3 pharmaceutics-17-00933-t003:** pH values of emulsion cream formulations before and after correction, and after one week of storage.

Sample	pH After Preparation	pH After Adjustment	pH After One Week
**E1**	7.50 ± 0.2	5.75 ± 0.2	6.25 ± 0.2
**E2**	7.40 ± 0.2	5.75 ± 0.2	5.85 ± 0.2 *
**E3**	7.20 ± 0.2 *	5.75 ± 0.2	5.75 ± 0.2 *

Values are expressed as mean ± standard deviation (*n* = 3). * indicates statistically significant difference compared to E1 (*p* < 0.05).

**Table 4 pharmaceutics-17-00933-t004:** Particle size percentiles (mean ± SD) of emulsion formulations.

Percentile	E1	E2	E3
D10 (μm)	4.68 ± 0.18	0.58 ± 0.004 *	0.50 ± 0.005 *
D50 (μm)	7.80 ± 0.25	0.78 ± 0.01 *	0.74 ± 0.01 *
D90 (μm)	17.10 ± 0.26	1.43 ± 0.01 *	1.49 ± 0.02 *

Values are presented as mean ± standard deviation (*n* = 3). * Indicates statistically significant difference compared to E1 (*p* < 0.05).

**Table 5 pharmaceutics-17-00933-t005:** Moisture content (%) of the emulsion-based cream formulations.

Sample ID	E1	E2	E3
Moisture Content (%)	90.35 ± 0.16	84.14 ± 12 *	79.41 ± 0.18 *

Values are presented as mean ± standard deviation (*n* = 3). * Indicates statistically significant difference compared to E1 (*p* < 0.05).

**Table 6 pharmaceutics-17-00933-t006:** Permeation flux of menthol and capsaicin in E2 formulation.

Compound	Flux (µg/cm^2^) Dermis	Flux (µg/cm^2^) Epidermis
Menthol	85.51 ± 16.64 *	9.85 ± 1.90 *
Capsaicin	3.40 ± 1.11	0.93 ± 0.19

Values are presented as mean ± standard deviation (*n* = 3). * Indicates statistically significant difference compared to capsaicin within the same skin layer (*p* < 0.05).

**Table 7 pharmaceutics-17-00933-t007:** Skin permeation flux (µg/cm^2^) of active compounds in the E3 formulation across dermis and epidermis layers.

Compound	Flux (µg/cm^2^) Dermis	Flux (µg/cm^2^) Epidermis
Menthol	86.41 ± 46.80 *	14.73 ± 5.64 *
Capsaicin	2.79 ± 1.44	1.18 ± 0.64
Glycine	142.79 ± 22.68 *	19.35 ± 7.77 *
Arginine	94.22 ± 11.88 *	34.17 ± 20.34 *
Histidine	99.59 ± 11.22 *	17.87 ± 11.16 *
Boswellic acid	0.78 ± 0.82	1.13 ± 0.95

Values are presented as mean ± standard deviation (*n* = 3). * indicates statistically significant difference compared to capsaicin within the same skin layer (*p* < 0.05).

## Data Availability

All data are available upon request.

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
