# Peer review of "Development and Evaluation of an Anti-Inflammatory Emulsion: Skin Penetration, Physicochemical Properties, and Fibroblast Viability Assessment"

_pharmaceutics, 2025, doi:10.3390/pharmaceutics17070933_

Round 1
Reviewer 1 Report
Comments and Suggestions for Authors
In this work, the authors have developed a novel topical formulation for skincare applications that can potentially manage inflammatory skin conditions.
The strengths of this manuscripts are as follows.
- The research is comprehensive. Selection of the control formulations is meaningful. Design of experiments and supporting the claims with experimental findings obtained from different instrumental techniques are appropriate.
- The topical formulation enriched with menthol, capsaicin, amino acids (glycine, arginine, histidine), and boswellic acid is novel. Its healing effects on inflammatory skin conditions has been found out to be positive.
- The references are relevant.
- Finally, the research seems to be appropriate to the article’s theme and can potentially attract MDPI’s audience.
Author Response
Comments 1: The research is comprehensive. Selection of the control formulations is meaningful. Design of experiments and supporting the claims with experimental findings obtained from different instrumental techniques are appropriate. The topical formulation enriched with menthol, capsaicin, amino acids (glycine, arginine, histidine), and boswellic acid is novel. Its healing effects on inflammatory skin conditions has been found out to be positive.
The references are relevant. Finally, the research seems to be appropriate to the article’s theme and can potentially attract MDPI’s audience.
The research is comprehensive. Selection of the control formulations is meaningful. Design of experiments and supporting the claims with experimental findings obtained from different instrumental techniques are appropriate.
The topical formulation enriched with menthol, capsaicin, amino acids (glycine, arginine, histidine), and boswellic acid is novel. Its healing effects on inflammatory skin conditions has been found out to be positive.
The references are relevant.
Finally, the research seems to be appropriate to the article’s theme and can potentially attract MDPI’s audience.
Response 1: We sincerely thank the reviewers for their encouraging and thoughtful feedback. We are pleased that the novelty of our formulation, the experimental design, and the relevance of our findings were appreciated. Your positive evaluation reinforces the importance of our work and motivates us to continue exploring innovative approaches for managing inflammatory skin conditions.
Reviewer 2 Report
Comments and Suggestions for Authors
The manuscript entitled "Development and Evaluation of an Anti-Inflammatory Emulsion: Skin Penetration, Physicochemical Properties, and Fibroblast Viability Assessment " presents some interesting results. However, in my view, the manuscript lacks crucial information throughout, which weakens the paper's overall impact. Several modifications are needed to enhance its quality.
- I think that Introduction can be reorganized due to some information are duplicated in different paragraphs. Also, the objective of this manuscript presented some reference. Please, revise it.
- In the Materials and Methods section, there are sentences (lines 84-90) that appear irrelevant to the manuscript. These should be carefully reviewed.
- It is missing some information about the suppliers of ingredients used to prepare the formulations. Were the formulations prepared in duplicate? The study focused on only 3 formulations, which seems a limited scope. Was this study a complementary study of previous paper?
- I suggest including some scheme of the production of emulsions.
- How were the concentration of ingredients chosen? What kind of recipients the emulsions were stored?
- There are some tips mistakes in the manuscript. Please, revise it.
- Regarding Equation 1, certain identifiers do not match the explanation provided for the equation. This should be corrected.
- The Materials and Methods section also lacks references and descriptions for some methods.
- How many replicates were performed for evaluating the pH and rheology?
- What cell line is used? Have all the formulations been tested? What were the controls? What concentrations were investigated?
- All the figures need to be improved. The quality was low, and some Figures (2 and 3) must be colored.
- What kind of statistical analysis is applied in the manuscript?
- Lines 319-325 represent duplicate information compared to method.
- Table 2 must be completed with spreadability, stickiness and adhesiveness results.
- The Discussion is missing comparisons with the literature.
- The Results section requires significant revision, as it contains descriptions of methods, duplicate information, and insufficient discussion.
Author Response
Comments 1: I think that Introduction can be reorganized due to some information are duplicated in different paragraphs. Also, the objective of this manuscript presented some reference. Please, revise it.
Response 1: Thank you very much for your helpful feedback. We have revised the Introduction to improve structure and clarity. Duplicate information was removed, and the objective is now stated clearly without a reference.
Comments 2: In the Materials and Methods section, there are sentences (lines 84-90) that appear irrelevant to the manuscript. These should be carefully reviewed.
Response 2: Thank you for your comments. Agree. We have reviewed the Material and Methods section.
Comments 3: It is missing some information about the suppliers of ingredients used to prepare the formulations. Were the formulations prepared in duplicate? The study focused on only 3 formulations, which seems a limited scope. Was this study a complementary study of previous paper?
Response 3: Thank you for your comments. Agree. We have now added detailed information about the origin and purity of all ingredients in the Materials and Methods section, including suppliers and analytical grade specifications.
Although only three final formulations (E1–E3) were presented in the study, several preliminary emulsions were initially developed and evaluated. These differed in their proportions of oleic acid and emulsifier and were screened based on their physicochemical properties, including visual stability, droplet size, viscosity, and centrifugation resistance. The three selected formulations demonstrated optimal stability, and we chose them for depth investigation.
All tested emulsions were prepared in duplicate to ensure reproducibility and consistency across evaluations.
The primary objective of this study was to evaluate the synergistic impact of capsaicin, menthol, and additional bioactive compounds (amino acids and boswellic acid) on both skin penetration and fibroblast viability under oxidative stress. Therefore, the selected formulations allowed us to isolate and compare the effects of capsaicin and menthol (E2) with the complete combination (E3), using E1 as the control. This design provided a targeted approach to studying the interactions of compounds within a native emulsion matrix.
This study is independent and not a direct continuation of a previous publication.
Comments 4: I suggest including some scheme of the production of emulsions.
Response 4: Thank you for the suggestion. Agree. We have now added Figure 2, which presents a schematic representation of the emulsion preparation process.
Comments 5: How were the concentration of ingredients chosen? What kind of recipients the emulsions were stored?
Response 5: Thank you for your comments. Agre. The concentrations of active ingredients were selected based on previously published literature and within the limits of commonly accepted safe and effective topical ranges. All emulsions were stored in plastic laboratory containers under controlled conditions to ensure sample stability throughout the analysis.
Comments 6: There are some tips mistakes in the manuscript. Please, revise it.
Response 6: Thank you for your comments. Agree.
Comments 7: Regarding Equation 1, certain identifiers do not match the explanation provided for the equation. This should be corrected.
Response 7: Thank you very much for your comments. We correct it.
Comments 8The Materials and Methods section also lacks references and descriptions for some methods.
Response 8: Thank you for your comments. Agree. We have revised the Materials and Methods section to include additional references and detailed descriptions for all key experimental procedures, including texture analysis, pH measurement, rheology, particle size determination, and skin permeation methods.
Comments 9: How many replicates were performed for evaluating the pH and rheology?
Response 9: Thank you very much for your comments. Agree. Both pH and rheological measurements were performed in triplicate for each emulsion sample, and the results were expressed as mean values with standard deviation.
Comments 10: What cell line is used? Have all the formulations been tested? What were the controls? What concentrations were investigated?
Response 10: Thank you for your comment. Agree. In this study, human dermal fibroblasts (HDFs) were used as the cell line to evaluate cytoprotective effects under oxidative stress conditions. All tested compounds (Boswellia extract, menthol, capsaicin, L-arginine, histidine, glycine) and their mixture were investigated individually. Controls included untreated cells (negative control) and cells treated with 300 µM H₂O₂ alone (positive oxidative stress control). Concentrations were selected based on preliminary screening and literature data: Boswellia extract and menthol at 20 µg/mL, capsaicin at 1–10 µg/mL, and amino acids at concentrations equivalent to those used in the emulsion formulation. The primary aim was to identify potential synergistic or individual cytoprotective effects of these compounds.
Comments 11. All the figures need to be improved. The quality was low, and some Figures (2 and 3) must be colored.
Response 11: Agree.Thank you for your feedback. We appreciate your observation. The image quality of all figures has been enhanced, and Figures 2 and 3 (now Figures 3 and 4) have been revised to include color for better visual clarity and readability. These updated figures have been incorporated into the revised manuscript.
Comments 12: What kind of statistical analysis is applied in the manuscript?
Response 12: Thank you for the observation. Statistical analysis was performed using one-way ANOVA followed by Tukey’s post hoc test to determine the significance between groups. Results are expressed as mean ± standard deviation (SD), and differences were considered statistically significant at p < 0.05. This information has now been clearly stated in the revised manuscript.
Comments 13:Lines 319-325 represent duplicate information compared to method.
Response 13: Thank you for your helpful comment. We’ve removed the repeated method details from this section and kept the focus on discussing the results instead. We hope this makes the section clearer and easier to follow.
Comments 14: Table 2 must be completed with spreadability, stickiness and adhesiveness results.
Response 14: Thank you for the suggestion. We have updated Table 2 to include the spreadability, stickiness, and adhesiveness results for all formulations.
Comments 15: The Discussion is missing comparisons with the literature.
Response 15: Thank you for your comment. Agree.The Discussion section has been revised to include relevant comparisons with recent literature to support our findings.
Comments 16: The Results section requires significant revision, as it contains descriptions of methods, duplicate information, and insufficient discussion.
Response 16: Thank you for your comment. Agree. We looked true all Results section to remove methodological descriptions, eliminate duplicate content, and integrate a clearer and more focused discussion alongside the results.
Reviewer 3 Report
Comments and Suggestions for Authors
This is very interesting and extensive research on development and evaluation of an anti-inflammatory emulsion containing active ingredients with significant skin permeation properties, and cytoprotective activity under oxidative stress conditions. Research design and methods were appropriate, results are clearly presented and text is well written. I suggest publication after some corrections, mainly related to minor text editing requirements listed below.
1) To additionally highlight significance of this study, it would be beneficial to add some comment into the section Introduction on previous availability of formulations which contain all these active ingredients and bioactive compounds or some of them in combination and what would be the main advance of this formulation in a relation to previously develop.
2) The origin and purity of used ingredients should be added in subsection Material and Methods.
3) Line 49, the sentence is incomplete, punctuation is missing and space is redundant.
4) On Figure 3, the y axis is missing.
5) What is the thermal and pH stability of used ingredients and bioactive compounds? Did the authors investigated these properties or used literature data?
Author Response
Comments 1: To additionally highlight significance of this study, it would be beneficial to add some comment into the section Introduction on previous availability of formulations which contain all these active ingredients and bioactive compounds or some of them in combination and what would be the main advance of this formulation in a relation to previously develop.
Response 1: Agree. Thank you very much for your helpful comments. We have now added a sentence in the Introduction highlighting that previous studies looked at some of these active ingredients individually or in pairs, but not all of them together. We also noted that our formulation is the first to combine these specific compounds and evaluate both skin penetration and protective effects on fibroblasts. Relevant references have been included to support this update.
Comments 2: The origin and purity of used ingredients should be added in subsection Material and Methods.
Response 2: Agree. Thank you for your comments. We have now added the origin and purity of all active ingredients and excipients in the Materials and Methods section.
Comments 3: Line 49, the sentence is incomplete, punctuation is missing and space is redundant.
Response 3: Agree. Thank you for your comments. The sentence on line 49 has been corrected.
Comments 4: On Figure 3, the y axis is missing.
Response 4: Thank you for your comments. Figure 3 has been corrected.
Comments 5: What is the thermal and pH stability of used ingredients and bioactive compounds? Did the authors investigated these properties or used literature data?
Response 5: Agree. Thank you for your comments. We did not perform separate thermal or pH stability studies for each active ingredient. However, literature data were taken into account during formulation development to ensure appropriate processing temperature and pH conditions for ingredient stability. In addition, the overall pH of the emulsion and its stability over time were experimentally evaluated (see Table 3), confirming that the final pH values remained within the optimal range for both ingredient integrity and skin compatibility.
Reviewer 4 Report
Comments and Suggestions for Authors
The authors have provided a comprehensive manuscript on the development and evaluation of an anti-inflammatory emulsion. This submission is worthy of publication in Pharmaceutics, and I only have a few minor comments that should be considered prior to acceptance.
- Line 384 -States that the initial pH values of all samples were pH -7.50 ± 10. Is this a typo?
- Line 545 – Did you perform any statistical tests to determine if the menthol flux for E3 is significantly greater than E2?
- Was there a reason why the marketed Capsaicin 0.075% cream was not used as control in all studies?
- In section 2.10, what was the extraction efficiency of L-Arginine, Glycine, Histidine, Capsaicin, Menthol and Boswellic Acid using the dry heat separation technique? Was the % extraction efficiency accounted for when determining the permeation flux in section 3.7?
- Did you conduct any recovery studies on the emulsion remaining on the skin after the ex vivo permeation study?
Author Response
Comments 1: Line 384 -States that the initial pH values of all samples were pH -7.50 ± 10. Is this a typo?
Response 1: Agree. Thank you for noticing this. Yes, this was a typographical error. The correct value is pH 7.50 ± 0.2. We have revised the text accordingly.
Comments 2: Line 545 – Did you perform any statistical tests to determine if the menthol flux for E3 is significantly greater than E2?
Response 2: Agree. Thank you for your comments. Statistical analysis was performed. The article was supplemented with a commentary on the comparison of E2 and E3.
Comments 3: Was there a reason why the marketed Capsaicin 0.075% cream was not used as control in all studies?
Response 3: Agree. Thank you for this important observation. The primary aim of our study was to evaluate the synergistic effects of multiple active compounds within a standardized emulsion base. Therefore, we used a self-prepared capsaicin-containing emulsion (E2) to ensure consistency in formulation parameters across all test groups. Commercial capsaicin creams often differ significantly in excipients, pH, and rheological properties, which could introduce variability and limit comparability in physicochemical and biological assessments. In contrast, the E2 emulsion, which contained only capsaicin (0.075%) and menthol (2%) within the same base as the other test formulations, was specifically developed to ensure consistency across all experimental groups. It served as an intermediate comparator to evaluate the effects of adding further bioactive compounds (amino acids and boswellic acid) in the full formulation (E3).
Comments 4: In section 2.10, what was the extraction efficiency of L-Arginine, Glycine, Histidine, Capsaicin, Menthol and Boswellic Acid using the dry heat separation technique? Was the % extraction efficiency accounted for when determining the permeation flux in section 3.7?
Response 4: Agree. Thank you for your comments. A dry heat method developed in-house is used to separate the skin layers (epidermis and dermis). The skin extraction method is developed in-house, and the solvent for each active ingredient is selected based on its solubility. The extraction efficiency was not evaluated in this study.
Comments 5: Did you conduct any recovery studies on the emulsion remaining on the skin after the ex vivo permeation study?
Response 5: Agree. Thank you for your comments. Although the study uses a pseudo-infinite dose, the amount of formulation is not large enough to perform recovery studies
Round 2
Reviewer 2 Report
Comments and Suggestions for Authors
I believe there was a mistake when sending the file for review. The authors sent the answers regarding the questions. However, the file sent is the same as before the review request. I have checked all the files and both v1 and v2 are the same document.
Please add the improved file for correct evaluation of the material.
Author Response

(The authors gave the same response as above.)

Round 3
Reviewer 2 Report
Comments and Suggestions for Authors
The manuscript entitled "Development and Evaluation of an Anti-Inflammatory Emulsion: Skin Penetration, Physicochemical Properties, and Fibroblast Viability Assessment " presents some interesting results and the authors improved the entire manuscript. However, in my view, a few modifications are necessary to complete the improvement, and, after that, the manuscript be accepted.
- I think the authors forgot to include a sentence about the use of menthol when rewriting the introduction. Furthermore, there is a species name that is not italicized.
- There are two Figures 2. Please, revise it.
- I suggest Table 2 be in landscape format to better present the results.
- Is there any study in literature that can be used to compare rheology results? I think this comparison with literature is important, in addition to the comparison between the formulations. What do the rheology results indicate?
Author Response
Comments 1: I think the authors forgot to include a sentence about the use of menthol when rewriting the introduction
Response 1: Thank you. Agree. We have now included a sentence regarding the potential role of menthol in the Introduction section (lines 63–66), highlighting its relevance as a TRP channel modulator and possible adjuvant in pain-relieving formulations. "In addition, compounds such as menthol have been explored for their topical analgesic effects via modulation of transient receptor potential (TRP) channels, suggesting potential value in combined pain-relieving formulations."
Comments 2: There are two Figures 2. Please, revise it.
Response 2: Thank you, agree. The figure numbering has been corrected accordingly — the duplicate Figure 2 has been renamed Figure 1, and all subsequent figures and their in-text references have been updated to ensure consistency throughout the manuscript.
Comments 3: I suggest Table 2 be in landscape format to better present the results.
Response 3: Thank you for your valuable suggestion. We have reformatted Table 2 into landscape orientation to improve the clarity and visual presentation of the data. The change is visible after removing Track Changes. We fully agree that, during the layout process, the table placement and orientation may be further adjusted by the journal's editorial team to achieve optimal visual presentation.
Comment 4: Is there any study in literature that can be used to compare rheology results? I think this comparison with literature is important, in addition to the comparison between the formulations. What do the rheology results indicate?
Response 4: Thank you for this important and insightful comment. We have addressed your suggestion by expanding the discussion in lines 394–404.